Identification of novel key markers that are induced during traumatic brain injury in mice

Li Yucheng 1
http://orcid.org/0009-0001-6101-4647 Li Ningbo 1
Luan Changjiao 1 2
Pei Yunlong 1
Zheng Qingbin 1
Yan Bingchun 3
Ma Xingjie 1 xingjie.ma@yzu.edu.cn
Liu Weili 1 zyyliuweili@126.com
1 Department of Intensive Care, the Affiliated Hospital of Yangzhou University, Yangzhou University , Yangzhou, Jiangsu , China
2 Department of Lung, the Third People’s Hospital of Yangzhou , Yangzhou , China
3 Department of Neurology, the Affiliated Hospital of Yangzhou University, Yangzhou University , Yangzhou , China
Orlov Yuriy
Electronic publication date: 2023 Aug 25
Publication date: 2023
Volume: 11
Electronic Location ID: e15981
Received 2023 May 1; Accepted 2023 Aug 8
Copyright: © 2023 Li et al.
Copyright year: 2023
Copyright holder: Li et al.
License: This is an open access article distributed under the terms of the Creative Commons Attribution License, which permits unrestricted use, distribution, reproduction and adaptation in any medium and for any purpose provided that it is properly attributed. For attribution, the original author(s), title, publication source (PeerJ) and either DOI or URL of the article must be cited.
License URL: https://creativecommons.org/licenses/by/4.0/

Keywords: Traumatic brain injury, ischemia/reperfusion, PBMCs, Cellular signaling

Funding: National Natural Science Foundation of China 32000509 Jiangsu University of Basic Science (Natural Science) Research Project 20KJB180003 Scientific Research Project of Jiangsu Commission of Health Z2020002 Social Development Project of Yangzhou Science and Technology Bureau YZ2020083 This work was supported by the support of the National Natural Science Foundation of China (32000509), the Jiangsu University of Basic Science (Natural Science) Research Project (No. 20KJB180003), and the Scientific Research Project of Jiangsu Commission of Health (Z2020002) to Xingjie Ma. Qingbin Zheng was supported by the Social Development Project of Yangzhou Science and Technology Bureau (YZ2020083). The funders had no role in study design, data collection and analysis, decision to publish, or preparation of the manuscript.

==============================
Background

Traumatic brain injury (TBI) has emerged as an increasing public health problem but has not been well studied, particularly the mechanisms of brain cellular behaviors during TBI.

Methods

In this study, we established an ischemia/reperfusion (I/R) brain injury mice model using transient middle cerebral artery occlusion (tMCAO) strategy. After then, RNA-sequencing of frontal lobes was performed to screen key inducers during TBI. To further verify the selected genes, we collected peripheral blood mononuclear cells (PBMCs) from TBI patients within 24 h who attended intensive care unit (ICU) in the Affiliated Hospital of Yangzhou University and analyzed the genes expression using RT-qPCR. Finally, the receiver operator characteristic (ROC) curves and co-expression with cellular senescence markers were applied to evaluate the predictive value of the genes.

Results

A total of six genes were screened out from the RNA-sequencing based on their novelty in TBI and implications in apoptosis and cellular senescence signaling. RT-qPCR analysis of PBMCs from patients showed the six genes were all up-regulated during TBI after comparing with healthy volunteers who attended the hospital for physical examination. The area under ROC (AUC) curves were all >0.7, and the co-expression scores of the six genes with senescence markers were all significantly positive. We thus identified TGM1, TGM2, ATF3, RCN3, ORAI1 and ITPR3 as novel key markers that are induced during TBI, and these markers may also serve as potential predictors for the progression of TBI.

Introduction

Traumatic brain injury (TBI) is defined as a pathological process of brain function changes or other manifestations caused by external forces. Briefly, the alterations in brain function caused by TBI can lead to one or more of the following clinical symptoms: (1) loss of memory for events immediately before (retrograde) or after the brain injury; (2) neurologic deficits (such as weakness, loss of balance, change in vision, dyspraxia, paresis, aphasia, etc.,); (3) loss or a decreased level of consciousness; (4) changes in mental state at the time when the injury occurs (disorientation, confusion, slowed thinking, etc.,) (Pervez, Kitagawa & Chang, 2018). TBI has been regarded as a worldwide leading cause of mortality in young adults and a main reason of death and disability across all ages, which cause a huge burden to families and society. However, the mechanisms as well as the prevention strategies for TBI remain to be further investigated and addressed.

Currently, the clinical treatment attentions towards TBI are mostly focused on refining neuro-intensive care protocolized therapies, establishing evidences for decompressive craniectomy, and developing novel pharmacological therapies (Khellaf, Khan & Helmy, 2019). Theoretically, the well-recognized pathophysiological mechanisms of TBI mainly include excitatory neurotoxicity, neuron oxidative stresses, neuron inflammation, and apoptosis etc., (Maas et al., 2017). Moreover, calcium accumulation induced apoptosis or cellular senescence in neurons has also been regarded as key players during TBI (Calì, Ottolini & Brini, 2012; Siesjö & Siesjö, 1996). In fact, intracellular calcium level is mainly controlled by sodium-calcium exchanger (NCX), transient receptor potential channels (TRP), and mitochondrial associated membranes (MAMs) etc., (Singh et al., 2019). For example, TRPV1 level is elevated after TBI, whereas inhibition of TRPV1 in vivo partially alleviates the TBI-induced blood brain barrier (BBB) disruption (Yang et al., 2019). Also, some calcium blocker related drugs (such as nimodipine) and calmodulin inhibitors have already been applied for clinical treatment of TBI (Singh et al., 2019).

Neuron apoptosis has been described as a common feature during TBI (Mira, Lira & Cerpa, 2021). However, little is known about the roles and functions of senescent neuron upon brain injury. Senescent cells have been known to secret chemokines and inflammatory factors via its senescence associated secretory phenotype (SASP), which are commonly triggered by nuclear factor kappa B (NF-κB) signaling, during development of age related diseases (Coppe et al., 2008). For example, senescent cells have been demonstrated to be accumulated in numerous neurologic disease, such as Alzheimer’s disease (AD) and Parkinson’s disease (PD) (He & Sharpless, 2017). Moreover, except for apoptotic neuron cells, massive secretion of inflammatory factors during TBI prompts us to ask if those “alive” neuron cells are senescent or not. If yes, can we try to unveil specific markers to recognize them? Based on these background, we were interested in investigating the expression profile of cellular senescence associated markers upon TBI.

Here, in this study, we performed a RNA-sequencing of frontal lobes on ischemia/reperfusion (I/R) mice and then verified the selected genes in peripheral blood mononuclear cells (PBMCs) of TBI patients as serum biomarkers have been demonstrated to be correlated with TBI clinical severity (Czeiter et al., 2020). Lastly, we applied receiver operator characteristic (ROC) curves and performed co-expression analysis with cellular senescence markers in brain noncancer samples to evaluate the predictive value of the interested genes.

Materials and Methods

Study design

The I/R mice model was established using transient middle cerebral artery occlusion (tMCAO) surgery, 3 days after, the brains were separated either for Triphenyl tetrazolium chloride (TTC) staining or for RNA-sequencing of frontal lobes. The blood samples from patients with TBI from the ICU in the Affiliated Hospital of Yangzhou University were separated for PBMCs and then prepared for reverse transcription quantitative polymerase chain reaction (RT-qPCR). Lastly, the ROC curves and the search-based exploration of expression compendium (SEEK) co-expression scores of the selected genes were performed.

Study population

As shown in Table 1, the inclusion criteria were as follows: patients with head injury in the ICU of the Affiliated Hospital of Yangzhou University; age >18 years; Glasgow Coma Scale (GCS) score ranges from 3 to 12. Patients with any of the following were excluded: combined with neurodegenerative diseases (Alzheimer’s disease, Parkinson’s, etc.), combined with tumors, combined with ischemic heart disease, combined with acute pancreatitis, combined with hydrocephalus. Our work has been carried out in accordance with The Code of Ethics of the World Medical Association (Declaration of Helsinki) and the Affiliated Hospital of Yangzhou University granted Ethical approval to carry out the study within its facilities (ethical application ref: 2020-YKL08-05). Verbal informed consent was obtained for experimentation with blood.

Table 1 The inclusion and exclusion criteria for this study.

Inclusion criteria	Exclusion criteria	
Patients with head injury in the
Department of Critical Care Medicine	Combined with neurodegenerative diseases	
Affiliated Hospital of Yangzhou University	(Alzheimer’s disease, Parkinson’s, etc.,)	
Age >18 years	Combined with tumors	
GCS score range from 3 to 12	Combined with ischemic heart disease	
	Combined with acute pancreatitis	
	Combined with hydrocephalus	

Study procedures

tMCAO surgery

ICR female mice were purchased from the Institute of Comparative Medicine of Yangzhou University. Mice were housed in a temperature-controlled, appropriate humidity animal facility with a cycle of 12 h light to 12 h dark. All mice had free access to water and commercial food. Mice following randomization were used to allocate experimental units to the indicated control or experimental groups. The tMCAO model was performed as previously described (Meng et al., 2021). Briefly, a total of 6 ICR female mice (30–35 g weight) (three for I/R group, three for sham group) were anesthetized with i.p. administration of sodium pentobarbital (50 mg/kg). 6/0 nylon sutures (Doccol Corporation, Sharon, MA, USA) with a silicone heat-rounded tip were inserted into the initiation site of the right middle cerebral artery (MCA) to block its blood flow under deep anesthesia. After 45 min, the filament was pulled out for blood reperfusion. The sham groups were operated same with tMCAO groups except filaments insertion. During the surgery and the recovery, the body temperature of mice was maintained at 37 ± 0.5 °C. 3 days later, mice were sacrificed through cervical vertebrae dislocation under deep anesthesia and the brains were dissected immediately and then were stored at −80 °C for further investigation. All studies were approved by the Science and Technology Commission of the Affiliated Hospital of Yangzhou University (ethical application ref: 2020-YKL08-05), and all protocols in this study were performed according to the Principles of Laboratory Animal Care (NIH publication No. 85Y50, revised 1996).

Assessment of infarct size (TTC staining)

Brains of I/R mice or mice in sham group were quickly separated after 3 days of surgery. The brains were sliced into four sections and stained with 2% TTC staining solution for 15–20 min. The normal area was stained with red, while the infarct area was white and the intersectional pink area means the penumbra.

RNA-sequencing

Total amounts and integrity of RNA from mice frontal lobes were assessed with the RNA Nano 6000 Assay Kit of the Bioanalyzer 2100 system (Agilent Technologies, Santa Clara, CA, USA). Then mRNA was purified from total RNA using poly-T oligo-attached magnetic beads. Fragmentations were carried out using divalent cations under elevated temperature in First Strand Synthesis Reaction Buffer (5X) and were used to build library. After qualification, the library was sequenced using the Illumina NovaSeq 6000 platform, and the end reading of 150 bp pairing was generated. Briefly, four fluorescent labeled dNTP, DNA polymerase and splice primers were added to the sequenced flowcell and were amplified. The sequencer captured the fluorescence signal of extensive chain and converted the optical signal into the sequencing peak, thus the sequence information of the tested fragments was obtained.

After sequencing, the index of the housekeeping genome was built and paired-end clean reads were aligned using HISAT2 (v2.0.5). FeatureCounts (v1.5.0-p3) was applied to count the reads numbers mapped to each gene. And then FPKM of each gene was calculated based on the length and reads count of the gene. Differential expression analysis was performed using the DESeq2 R package (1.20.0). Also, the gene set enrichment analysis (GSEA) tool (http://www.broadinstitute.org/gsea/index.jsp) was used to analyze the Kyoto Encyclopedia of Genes and Genomes (KEGG) and Gene Ontology (GO) enrichment.

PBMCs isolation

Peripheral blood samples were collected from either brain injury patients within 24 h since arriving ICU or healthy volunteers who were at the hospital for a physical examination. PBMC donors gave informed consent, and all studies were approved by the Science and Technology Commission of the Affiliated Hospital of Yangzhou University. PBMCs were isolated within 4 h using a human peripheral blood lymphocyte isolation kit (LTS10771, TBD, China) according to the manufacturer’s instructions. The separated PBMCs were added to TRNzol (TIANGEN) and stored at −20 °C until using.

RNA extraction, reverse transcription, and real-time quantitative PCR

TRNzol (TIANGEN) were used to extract total RNA from PBMCs according to the manufacturer’s instructions. After then, the PrimeScript™ RT Master Mix (TAKARA) kit was applied to synthesize cDNA from 500 ng RNA following the manufacturer’s recommendations, and then the reverse transcription (RT) reaction mix was used as templates after diluting 1:10 times for quantitative PCR (qPCR) analysis. The total of 10 μL mixture of qPCR reaction contained Hieff® qPCR SYBR Green Master Mix (No Rox) (YEASEN; Shanghai, China), 400 nM of both forward and reverse primers, and 1.5 μL of cDNA. Primer sequences are listed in Table 2. The qPCR analyses were performed using a cFX96 Touch Thermocycler (Bio-Rad, Hercules, CA,USA) with following program: 95 °C 5 min, followed by 40 cycles of 95 °C 10 s, 60 °C 30 s, and end with a melting curve from 65 °C to 95 °C with increments of 0.5 °C 5 s. The PCR reactions were performed with technical duplications. The relative expression of indicated mRNA level was calculated using the comparative Ct (2−ΔΔCt) method after normalized to the mean of PGK1 and HPRT1 housekeeping genes.

Table 2 Human primers used for RT-qPCR in this study.

Gene symbol	Forward primers	Reverse primers	
PGK1	GCCAAGTCGGTAGTCCTTATG	CCCAGCAGAGATTTGAGTTCTA	
HPRT1	CGAGATGTGATGAAGGAGATGG	TTGATGTAATCCAGCAGGTCAG	
TGM1	TGCTGGATGCCTGCTTATAC	TTCACCATGGCAGAGATGAC	
TGM2	TGTTGGTCAGAGGAGTGATTG	GGAGTGGACCTTGTGGTTATT	
ATF3	CTGGAAAGTGTGAATGCTGAAC	ATTCTGAGCCCGGACAATAC	
RCN3	GCTGTCATAGTCCCAGAGGATA	GGTTCTTAGAGCTGAGCTTGG	
ORAI1	CCCTTCGGCCTGATCTTTATC	GGAACTGTCGGTCAGTCTTATG	
ITPR3	CAGCTCTCCAGGCACAATAA	GGCTGAGCATGGAAGAGATAC	
NF-Kb1	CTCCACAAGGCAGCAAATAGA	ACTGGTCAGAGACTCGGTAAA	
RELA	TGAGCCCACAAAGCCTTATC	ACAATGCCAGTGCCATACA	

Statistics

GraphPad Prism 9 software was used for the statistical analyses. Two-tailed unpaired Student’s t-test were applied to determine the P value (ns, non-significant; *P < 0.05, **P < 0.01, ***P < 0.001, and ****P < 0.0001). D’Agostino & Pearson’s normality test was used before analyses to evaluate the samples normality.

Results

Screen key genes that are induced during TBI

The I/R brain injury mice model using tMCAO strategy was performed and the infarct size was shown to evaluate the successful establishment of TBI mice (Fig. 1A). Then the frontal lobes of I/R brain injury mice were sent for RNA-sequencing. The pearson correlation between samples were shown to confirm the homogeneity of samples in both groups (Fig. 1B). A total of 22,146 genes were enrolled, among which 3,475 genes were found to be up-regulated and 2,958 genes were down-regulated according to the sequencing data (Fig. 1C). Among the up-regulated genes, the apoptosis and NF-κB signaling were found to be significantly activated based on the KEGG analysis (Fig. 1D). Interestingly, the GSEA also showed an activation of cellular senescence signaling during TBI (Fig. 1E). Finally, based on the novelty in term of TBI regulation and implication in cellular apoptosis or senescence, as well as the fact that TBI induces neuron cells calcium overload (Mira, Quintanilla & Cerpa, 2023), we were interested in Transglutaminase 1(TGM1), TGM2, and calcium related genes activating transcription factor 3 (ATF3), Reticulocalbin 3 (RCN3), ORAI calcium release-activated calcium modulator 1 (ORAI1), and inositol 1,4, 5-trisphosphate receptor type 3 (ITPR3) (Table 3).

Figure 1 Analyses of RNA-sequencing of frontal lobes from I/R mice.

The I/R brain injury mice model using tMCAO strategy was performed and the infarct size was evaluated by TTC stained (A). A total of 6 mice were sent for RNA-sequencing (three for each group). The pearson correlation between samples (B), the volcano plot of different expressed genes (DeGs) (C), the KEGG analysis of up-regulated genes (D), the GSEA analysis of cellular senescence signaling (E) were shown.

Table 3 Summary of the demographic characteristics of the study cohort.

Syobol	I_R_1	I_R_2	I_R_3	sham_1	sham_2	sham_3	I_R	sham	Fold change	P value	
TGM1	2,053.759	2,329.378	1,810.78	0	4.67818	7.089068	2,064.639	3.922416	9.015045	7.92E−81	
TGM2	3,647.751	2,455.565	3,754.375	315.7232	324.6657	210.6466	3,285.897	283.6785	3.53381	2.24E−41	
ATF3	2,491.96	1,916.216	1,334.702	28.60579	14.97017	8.101792	1,914.293	17.22592	6.803879	1.78E−64	
RCN3	186	275	259	113	86	94	234.8125	98.46042	1.25557	1.33E−05	
ORAI1	311	417	480	186	217	197	392.7291	199.8672	0.974492	2.66E−05	
ITPR3	272	360	705	127	149	161	430.7989	145.6705	1.564671	9.80E−06	

Validation of the key genes in PBMCs of TBI patients

To verify the up-regulation of the six genes from RNA-sequencing data, we then tested the blood samples from TBI patients. A total of 12 TBI patients met the included criteria. Ages were ranged from 37 to 77 years old, GCS scores range from 3 to 12, APACHEII scores range from 15 to 34 (Table 4).

Table 4 Summary of the demographic characteristics of the study cohort (12 specimens).

	Age	Gender	Diagnosis	GCS scores	APECHEII scores	D dimer	Base excess	Hemoglobin	Leucocyte	C-reactive protein	
1	40	Male	Cerebral hemorrhage	5	22	0.51	−3.7	167	23.65	4.55	
2	55	Male	Traumatic subarachnoid hemorrhage	8	20	7.7	−4.1	134	10.95	1.81	
3	69	Female	Ventricular hemorrhage	5	19	2.51	−2.4	129	7.23	0.56	
4	55	Male	Severe brain injury	6	17	95.22	−4.3	96	17.08	1.04	
5	42	Male	Severe brain injury	3	20	8.09	4.3	79	21.98	156.56	
6	53	Female	Ischemic hypoxic encephalopathy	3	21	4.45	4	111	11.56	8.92	
7	68	Male	Cerebral contusion	6	26	66.91	0.1	124	16.01	0.2	
8	65	Male	Traumatic subarachnoid hemorrhage	11	34	31.79	−2.4	132	15.61	8.72	
9	46	Male	Basal ganglia hemorrhage	13	15	21.56	−1.7	0.12	8.78	2	
10	77	Female	Cerebral hemorrhage	9	19	0.726	3.5	110	10.93	0.4	
11	76	Male	Brainstem infarction	13	25	0.84	−7.5	126	11.73	2.51	
12	37	Male	Brainstem hemorrhage	6	22	0.27	−1.3	146	10.57	6.38	
Hematocrit (%)	Systolic blood pressure	Diastolic blood pressure	pH	PaO2	PaCO2	HCO3−	Hypertension	Diabetes	
50	220	111	7.43	72	30	19.3	YES	NO	
39.2	138	72	7.31	162	44	22.2	NO	NO	
129	166	100	7.38	76	38	22.3	YES	YES	
27.9	131	56	7.32	111	42	21.6	NO	NO	
24	97	62	7.56	114	30	26.6	NO	NO	
35.1	136	61	7.49	169	36	27.4	NO	NO	
37.1	104	62	7.25	381	66	28.9	YES	YES	
37.6	160	90	7.42	126	33	23.1	YES	YES	
45.2	140	80	7.39	114	38	23	YES	NO	
32.8	189	43	7.42	184	44	27.7	YES	NO	
38.6	156	90	7.4	105	26	16.1	YES	NO	
43.1	129	78	7.36	103	43	24.3	YES	NO	

We then collected blood samples from healthy volunteers as control to normalize the expression of the six genes in the 12 TBI patients. As expected, the expression levels of TGM1, TGM2, ATF3, RCN3, ORAI1 and ITPR3 in TBI patients were all significantly elevated after comparing with control group (Figs. 2A–2F). As most of the secreted inflammatory factors during brain injury are mediated by NF-κB signaling (Hoare et al., 2016), we then tested the expression of NF-κB subunits NF-κB1 and RELA, the results gave the same as the six genes that NF-κB1 and RELA were both induced during TBI (Figs. 2G and 2H). These data indicate that the expression of the screened six genes were truly stimulated during TBI. These observations also illustrated that calcium is overloaded after TBI.

Figure 2 RT-qPCR analyses of indicated genes from PBMCs of TBI patients.

RNA was extracted from PBMCs of 12 TBI patients and 12 healthy volunteers. RT-qPCR was performed to analysis the relative expression of TGM1 (A), TGM2 (B), ATF3 (C), RCN3 (D), ORAI1 (E), ITPR3 (F), NF-κB1 (G) and RELA (H) (*P < 0.05, ***P < 0.001, and ****P < 0.0001).

ROC curves of the key genes and their co-expression with senescence markers

To further assess the predictive value of the six genes for TBI progression, we established ROC curves for each gene (Fig. 3). The AUCs corresponding to TGM1, TGM2, ATF3, RCN3, ORAI1, and ITPR3 were 0.972, 0.951, 0.986, 0.962, 0.729, and 0.896, respectively, indicating the expression of the six genes is meaningful for the prediction of TBI progression. Besides, the AUCs of NF-κB1 and RELA were 0.979 and 0.764, which means the inflammation occurs during TBI (Table 5).

Figure 3 The ROC curves and AUCs of the six genes.

Table 5 AUCs of TGM1, TGM2, ATF3, RCN3, ORAI1, ITPR3, NF-κB1 and RELA in TBI patients.

Test result variables	Area	Standard errora	Asymptotic prominenceb	Asymptotic 95% confidence interval	
Lower limit	Upper limit	
TGM1	0.972	0.031	0.000	0.912	1.000	
TGM2	0.951	0.049	0.000	0.855	1.000	
ATF3	0.986	0.018	0.000	0.950	1.000	
RCN3	0.962	0.038	0.000	0.888	1.000	
ORAI1	0.729	0.117	0.057	0.499	0.959	
ITPR3	0.896	0.066	0.001	0.767	1.000	
NF-KB1	0.979	0.023	0.000	0.935	1.000	
RELA	0.764	0.103	0.028	0.561	0.966	
Notes:

a Assumed by nonparametric values.

b Null hypothesis: True region = 0.5.

As senescent cells were found to be accumulated in neurodegenerative diseases (Musi et al., 2018), we then asked if the six genes could be relevance with senescence markers in brain cells. We searched the SEEK co-expression database and found out that the six genes were among the genes which expression are significantly positively correlated with both cyclin dependent kinase inhibitor 2A (CDKN2A) and growth differentiation factor 15 (GDF15) senescence markers in brain noncancer samples from 81 datasets (Fig. 4). These data indicate that neuron cells after TBI may suffer from cellular senescence, and, more importantly, might accompanied with calcium overload. Altogether, these observations support that the identified six genes are good predictors for TBI progression.

Figure 4 (A–F) SEEK co-expression analysis of the six genes with senescence markers CDKN2A and GDF15.

The SEEK co-expression database was interrogated for genes co-expressed with (A) TGM1, (B) TGM2, (C) ATF3, (D) RCN3, (E) ORAI1, and (F) ITPR3 in brain (noncancer) and the co-expression information of CDKN2A and GDF15 with the indicated genes were shown.

Discussion

In this study, we performed a RNA-sequencing of frontal lobes from I/R mice, and this approach allowed us to screen out six genes named TGM1, TGM2, ATF3, RCN3, ORAI1, and ITPR3 which are significantly up-regulated during TBI. After then, the six genes were further verified in the PBMCs of clinical TBI patients. Additionally, the corresponding ROC curves and co-expression analysis support that the six genes may be potential diagnostic or predictors for TBI progression. Our study at least has the following significances: (1) is conducive for the prognosis of TBI; (2) is beneficial to enrich and offers a deeper understanding of the pathophysiological mechanism of TBI; (3) presents novel potential routes for further study of TBI.

Multiple injury models have been demonstrated to trigger TBI, including fluid percussion injury (FPI) (Rowe, Griffiths & Lifshitz, 2016), controlled cortical impact (CCI) (Siebold, Obenaus & Goyal, 2018) and tMCAO (Liu et al., 2020). To simulate replicative TBI models, the animal models, including mice, rats, and rabbits, have been widely introduced and investigated. These approaches have identified multiple novel TBI markers such as tumor necrosis factor receptor associated factor 2 (TRAF2) (Li et al., 2019), phosphorylated connexin 43 (p-Cx43) (Chen et al., 2018), and heat shock protein 70 (Hsp70) (Yuan et al., 2020) and extended our understanding towards the secondary injury after TBI.

In addition, numerous studies have focused on the mechanisms and development of neuron injury during TBI. Calcium overloading, for example, has been widely known for its function of inducing apoptosis of neurons (Huang et al., 2017). Here, in our study, we identified several calcium associated genes (ATF3, RCN3, ORAI1, and ITPR3) that were strikingly induced after TBI. ATF3 is described as a transcription factor and is involved in the complex processes of cellular stress responses via binding to specific target genes (Lee et al., 1987). For instances, macrophages ATF3 plays key roles in host survival and pathogen clearance through controlling intracellular calcium level and ATP homeostasis, and consequently stimulating interleukin 17A (IL-17A) expression during streptococcus pneumoniae infection (Lee et al., 2018). ATF3 was also found to be induced in myocardial ischemia and myocardial ischemia coupled with reperfusion and renal I/R (Fan & Yang, 2017). The endoplasmic reticulum (ER) calcium binding protein RCN3 is known to maintain the intracellular calcium homeostasis (Cai et al., 2022). RCN3 has also been found to regulate ovarian follicle development, participate in perinatal lung maturation, and regulate fibrillogenesis (Jin et al., 2016, 2018). ORAI1 is membrane calcium channel subunit that is activated by the calcium sensor stromal interaction molecule 1 (STIM1). Normally, STIM1 senses the ER-calcium depletion, for example in astrocytes, and subsequently binds ORAI1 to activate intracellular calcium influx (Moreno et al., 2012). The ER located ITPR3 is implicated in ER calcium release upon stresses. As shown, elevated ITPR3 expression in neurons contributes to the increasing of intracellular calcium level, which in turn leads to calcium overload induced cell dysfunction such as neuronal cytotoxicity, neuroinflammation, or oxidative stresses (Blackshaw et al., 2000). TGM1 and TGM2 are both enzymes that regulate protein synthesis, and overexpression of TGM1 was found to be toxic for neurons (Tripathy et al., 2020).

Senescent cells exhibit profound alterations in their transcriptomes, resulting in a complex pro-inflammatory responses which constitute the SASP. Studies demonstrated that activation of pro-inflammatory markers in neurons and glial cells after mild TBI in mice may lead to long-term neurological symptoms and risk of neurodegenerative disease, while cleanness of those senescent cells using senolytics significantly improved performance in morris water maze (MWM) (Schwab et al., 2022). Here we found the SASP mediators NF-κB1 and RELA were both induced after TBI. We thus speculate that a number of neurons are senescent during TBI and the SASP factors secreted by these cells not only induce inflammation but also accelerate the senescence phenotype of neighbor cells in a vicious circle. However, the percentage and the exact roles of the senescent neuron cells during TBI remain partially understood, further investigations towards the function of these six genes in the senescent neuron during TBI need to be done. In addition, as cellular senescence may also play vital roles in wounding healing and tumor suppression, the exact burden of the senescent cells in tissues need to be carefully investigated in the future.

However, after performing the inclusion criteria, only 12 TBI patients were enrolled in this study. This makes it difficult to draw certain conclusions and is likely subject to selection bias. There are also differences in the judgment of GCS scores as they were from subjective assessment, which may bring deviations for the evaluation of prognosis. Moreover, whether or not the identified six genes are truly implicated or specifically induced by TBI remains unclear, thus further studies towards the function and roles of these genes in TBI need to be carefully investigated. Furthermore, the analyses are limited due to the single-center study. Therefore, the results should be interpreted with some caution.

Conclusions

This study presents novel TBI markers termed TGM1, TGM2, ATF3, RCN3, ORAI1, and ITPR3 based on a RNA-sequencing of frontal lobes from I/R mice and verifications in clinical TBI patients. Then, the clinical prediction or diagnostic values of these genes were predicted using ROC curves and co-expression strategies. Challenges ahead will be to investigate the roles and function of TGM1, TGM2, ATF3, RCN3, ORAI1, and ITPR3 in TBI as well as TBI-induced secondary brain disorders.

Supplemental Information

Supplemental Information 1 Author checklist.

Click here for additional data file.

Supplemental Information 2 Raw data for RNA-sequencing.

Click here for additional data file.

Supplemental Information 3 Raw data for RT-qPCR.

Click here for additional data file.

Supplemental Information 4 Raw data for SEEK analysis.

Click here for additional data file.

Supplemental Information 5 RNA-sequencing data - all genes comparation.

Click here for additional data file.

Supplemental Information 6 RNA-sequencing data - all sequences.

Click here for additional data file.

Supplemental Information 7 Raw data for Table 4.

Click here for additional data file.

Additional Information and Declarations

Competing Interests

Author Contributions

Human Ethics

Animal Ethics

Data Availability

The authors declare that they have no competing interests.

Yucheng Li performed the experiments, analyzed the data, prepared figures and/or tables, authored or reviewed drafts of the article, and approved the final draft.

Ningbo Li performed the experiments, analyzed the data, prepared figures and/or tables, and approved the final draft.

Changjiao Luan performed the experiments, analyzed the data, prepared figures and/or tables, and approved the final draft.

Yunlong Pei performed the experiments, analyzed the data, authored or reviewed drafts of the article, and approved the final draft.

Qingbin Zheng performed the experiments, prepared figures and/or tables, and approved the final draft.

Bingchun Yan performed the experiments, prepared figures and/or tables, and approved the final draft.

Xingjie Ma conceived and designed the experiments, performed the experiments, prepared figures and/or tables, authored or reviewed drafts of the article, and approved the final draft.

Weili Liu conceived and designed the experiments, prepared figures and/or tables, authored or reviewed drafts of the article, and approved the final draft.

The following information was supplied relating to ethical approvals (i.e., approving body and any reference numbers):

The Affiliated Hospital of Yangzhou University granted Ethical approval to carry out the study within its facilities (Ethical Application Ref: 2020-YKL08-05).

The following information was supplied relating to ethical approvals (i.e., approving body and any reference numbers):

The Science and Technology Commission of the Affiliated Hospital of Yangzhou University provided full approval for this research (2020-YKL08-05).

The following information was supplied regarding data availability:

The raw measurements are available in the Supplemental Files.

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
