# Peer review of "Identification of novel key markers that are induced during traumatic brain injury in mice"

_PeerJ, doi:10.7717/peerj.15981_

## Round 0.1 · original submission · Minor Revisions

We have received two reviews suggesting minor revision. Third review is pending. Way send additional comments later in order not delay the paper production process. Overall, there are no critical remarks.

The topic of traumatic brain injury study is interesting. I’d suggest extending the discussion section to mention brain injury marker studies on laboratory animals. There are some works in mice, including gene expression changes. At the authors’ discretion, the figures could be all in color.

Figure 1 (panel f) could be presented as a separate table. Same remark is for Figure 3 panel b.

Figure 4 is in black color. May keep it as is, but please make larger font size for all the figures.

Reviewer 1 ·

Basic reporting

The manuscript is well written, with results clearly described.

Experimental design

The methods are well written, with experimental design reported in detail.
- Animal size is not mentioned until the figures, please check.

Validity of the findings

There are a few questions that need to be addressed before the manuscript can be considered to be published.

1) While the tMCAO strategy was used to induce TBI, do other animal models such as fluid percussion injury or cortical impact injury show similar results?

2) The results seem to be very particularly related to senescence in neurons, which is not particular to TBI, but broader property of apoptosis induced changes.

3) The authors selected 6 genes as markers which were differentially regulated with the reason stated as "due to their novelty in term of TBI regulation and apoptosis or senescence regulation". The authors must discuss why these 6 genes are selected out of 3475 genes, and why the other genes were not selected. Is it statistical threshholding, or some other mathematical reasoning behind this? Are other genes not biomarkers for TBI?

4) The selected 6 genes are also differentially regulated in the PBMC of TBI patients, but are those specific to TBI patients, or across all patients with neuron damage? I am unsure, and this goes back to point 2, whether these genes are specifically regulated with TBI or overall neuron apoptosis?

The arguments while well presented, need refinement with additional reasoning before the manuscript is accepted.

Reviewer 2 ·

Basic reporting

The article is well structured, and follows a logical layout and progression. The introduction clearly lays out the rationale for the work, this being that further understanding of the molecular basis of traumatic brain injury (TBI) is needed to improve disease management and therapeutic interventions. The introduction would benefit from more clearly and further outlining the rationale for investigating the links between TBI and neuronal senescence/cellular senescence associated markers i.e. why focus on these markers in TBI. The figures are well laid-out and presented and clear to interpret. All of the figures, tables and supplemental materials referred to in the manuscript are included in the submission. The results and findings are generally clearly described in the text, but the results section of the manuscript could be improved by further expanding on the underlying rationale for and description of the findings. The raw data associated with the findings is supplied, but would benefit from improved labelling and description for easier understanding by others. As currently it is not clear in what form the raw data is supplied or which results it relates to, although presumably it is dCt values from the qPCR shown in Figure 2. The two subsections in the Results, ‘Patients characteristics’ and ‘Validation of the key genes in PBMCs’ would benefit from being merged to comprise a more complete and coherent larger subsection. The English language used is generally comprehensible throughout, but would benefit from further proofreading and modifications in multiple instances. In particular, I have a feeling that the authors may have intended to use the word ‘predictor’ rather than ‘predicator’, which is written multiple times throughout the manuscript, for example, line 33 – “predicative value of the genes” or line 41 “potential predicators for the progression of TBI”.

Experimental design

The research question and rationale of the study is clearly defined in the manuscript, which is to gain improved understanding of the molecular basis and mechanisms that underlie TBI disease presentation and progression. This is a very relevant question to address, as this is a topic that needs improved understanding and research in this area could lead to improvements in the diagnosis, management and treatment of TBI related disease which is a serious cause of mortality and disability. As stated above, the links between TBI and neuronal senescence stated in the paper could be more clearly defined in the manuscript. The authors utilize a common experimental mouse model of TBIs in the form of an ischemia/reperfusion (I/R) brain injury model using transient middle cerebral artery occlusion (tMCAO). The authors then utilize RNA-sequencing to investigate gene expression changes at the transcriptional level in the brains of ischemia/reperfusion (I/R) brain injury compared to control mice. Bulk RNA-sequencing is a highly relevant methodological approach to analyse global changes in gene expression that occur during TBI, and to identify individual genes / markers with novel roles in the disease. The authors identify a subset of genes with roles in neuronal senescence to be activated during TBI. The clinical relevance of these targets was further validated by confirming upregulated expression using an rt-qPCR approach in PBMCs isolated from blood samples from TBI patients compared to health volunteers. The methods are generally well detailed and contain a sufficient level of detail for the experiments to be reproduced by other researchers. However, no detail is given in the methods regarding the RNA-sequencing approach used. The authors should provide information on the sequencing in terms of which fraction of cellular RNA was sequenced (e.g. polyA selected or ribo-depleted?) sequencing platform and sequencing parameters used to generate the data, as well as which software package, version, parameters and workflow were used in the downstream processing and analysis of the data to generate the findings e.g. information on read alignment, the differential gene expression analysis, and Gene Set Enrichment Analysis. It would also be of great benefit for the data to be made available in a public repository e.g. the NIH Gene Expression Omnibus (GEO) repository or similar, as it is not clear that this has been done, and to provide accession numbers/IDs for the public dataset. The study upholds appropriate ethical standard for the mouse and human studies performed. The manuscript states that the animal and patient studies were approved by relevant institutional ethical review boards, and ethical approval review forms are provided.

Validity of the findings

The major conclusions of the paper are identifying gene differentially expressed in the brain during TBI, and validating these using patient/healthy volunteer human blood samples. The conclusion in the discussion section on line 206 that the study, “provides novel potential therapeutic targets for clinical treatment of TBI” is perhaps too strong, as no mechanistic role for any of the identified targets in the pathogenesis of TBI is presented. It would be better to state that these identified genes present routes for further research. The data is well controlled and use of statistics in the paper is sound, for example, student’s t-test for pair-wise comparisons. However, the RNA-sequencing comprises a major part of the manuscript, and further detail on the downstream processing and statistical analysis of the RNA-sequencing data should be provided. As also previously stated, the RNA-sequencing dataset should be made publicly available in a data repository, and dataset accession/ID numbers quoted in the methods of the paper to link to the corresponding dataset.

Additional comments

The authors seek to characterise the molecular basis of events underpinning traumatic brain injury (TBI). TBI is a major cause of both mortality and morbidity, and therefore research to further basic understanding of the cellular and molecular basis of this medical phenomenon is highly relevant and warranted in order to inform future approaches aimed to improve pathology management and develop improved therapeutic interventions. The work presented here comprises a small study to investigate the molecular underpinnings of TBI. The authors use a common experimental mouse model of TBIs such as ischemic stroke in the form of an ischemia/reperfusion (I/R) brain injury model using a transient middle cerebral artery occlusion (tMCAO) strategy. The gene expression changes and profiles in cells of the frontal lobe brain region were then investigated at the transcriptional level by RNA-sequencing, comparing frontal lobes tMCAO and control mice. The RNA-seq approach identified novel genes upregulated during TBI, that may function as markers of brain injury induction or progress. The clinical relevance of the induction of these identified genes was further confirmed and validated by analysing and comparing gene expression by rt-qPCR in PBMCs isolated from blood samples from healthy volunteer donors and TBI patients. This confirms the relevance of these genes identified from RNA-seq in human disease. Overall this study identifies multiple upregulated genes and markers that are novel for their role in TBI, and which may serve as useful prognostic markers for the presence or progression of TBI in patients, or as targets for further investigation into the molecular basis of disease. The RNA sequencing dataset may be a useful resource for others working in the field.

---

## Round 0.2 · Minor Revisions

Please check the remaining comments from Reviewer #1 about generalization of biomarkers for TBI. Change paper title to be more focused (on mice model only). Extend discussion part. The remaining remarks are really minor, but should be fixed.

Reviewer 1 ·

Basic reporting

The report is well presented

Experimental design

The experimental design is well presented

Validity of the findings

The major question asked previously was whether the key biomarkers showed in this article were specific to TBI induced senescence or whether it was generalised cell apoptosis, to which the authors replied they are not sure and this needs to be well studied.
Given the article states that the authors identified key biomarkers for TBI (but unsure if they are generalizable to other injuries/ cell death in the brain), they should either modify the title that truly reflects their work or mention limitations of this study in the discussion section. If the authors had showed similar results in other mouse models of TBI, their case would have been more convincing.

Other concerns were responded to well.

Reviewer 2 ·

Basic reporting

No comment

Experimental design

No comment

Validity of the findings

No comment

Additional comments

I commend the authors on the changes made. These amendments greatly improve the manuscript.

---

## Round 0.3 · accepted · Accept

Thank you for the manuscript update. There are no more remaining comments on this work.